# Dear student, what should I write on my wall? A case study on academic uses of Facebook and Instagram during the pandemic

**Claudiu Coman** [1]*, **Luiza Mesesan-Schmitz**[1], **Laurentiu Gabriel Tiru**[2], **Gabriela Grosseck**[3], **Maria Cristina Bularca**[1]

**1** Department of Social Sciences and Communication, Faculty of Sociology and Communication, Transilvania University of Brasov, Brasov, Romania, **2** Department of Sociology, Faculty of Sociology and Psychology, West University of Timisoara, Timisoara, Romania, **3** Department of Psychology, Faculty of Sociology and Psychology, West University of Timisoara, Timisoara, Romania

* claudiu.coman@unitbv.ro

**Data Availability Statement:** All relevant data are within the manuscript and its Supporting Information files.

**Funding:** The authors received no specific funding for this work.

## Abstract

Prior to the COVID 19 pandemic, discussions about online learning referred to the use of e-learning platforms and social networks as auxiliary tools in the educational process. Due to the pandemic, universities were forced to adopt an exclusive online teaching process and most universities today use platforms dedicated to online learning such as Moodle platforms. In this context, we were interested in analyzing the attitude of students regarding the way social networks could be integrated into the educational process, and if the positive attitude of students towards social networks and their use for academic purposes, proven in previous studies, remains positive under the conditions generated by the pandemic. In this regard, the present study aimed at identifying the attitude of Romanian students towards the use of Facebook and Instagram as educational tools and the circumstances in which students believe these platforms could be used by them and their teachers. An online survey was conducted on 872 students from public higher education institutions in Romania. Based on the exploratory factor analysis and the parametric test, the empirical results show that students have a slightly positive attitude towards using Facebook in the educational process, but they have a more reticent, less positive attitude towards using Instagram. Thus, the most appropriate contexts in which these platforms could be used are represented by extracurricular activities. A higher preference for the use of Facebook rather than Instagram, was identified among master and PhD students. No major differences were revealed in student subgroups sorted by gender or study domain.

## Introduction

In the age of technology, the online environment and its fast evolution has managed to eradicate the barriers of physical distance in the communication process, facilitating the development of new methods of sharing and receiving information. With the explosion of internet usage a new virtual medium of interaction and socialization had developed [1]. From the

**Competing interests:** The authors have declared that no competing interests exist.

perspective of the current generations of students, the online environment is highly appreciated, most students calling upon web sources when they are asked to complete documenting activities, bibliographic resources being consulted only after searching the internet [2]. Studies reveal that social networks are important in the life of university students, platforms such as Facebook being considered by students appropriate for supplementing the traditional face—to—face courses, and also being perceived as platforms that can improve the quality of the courses and which can help them better communicate with their peers and feel more connected to them [3].Thus, the efficiency of using social networks as educational tools in higher education, was proven in many universities [4–7].

Although E-learning platforms or social networks were used by higher education institutions in the educational process prior to the COVID-19 pandemic, due to the pandemic, universities were forced to adapt the educational process for exclusively online learning, most of them opting for the integration of E-learning platforms such as Moodle platforms.

Recent studies that focused on identifying the perception of Romanian students about online learning during the pandemic [8, 9], revealed less positive attitudes towards this type of learning, students considering that the online educational process has less value than the traditional one, preferring instead to use E-learning platforms as a complementary method to traditional, face-to-face learning. Their attitude can be justified by a series of demotivating factors that can affect their academic performance, such as: technical issues, teachers' lack of technical skills, poor communication with teachers, or the maladjustment of the teaching style to the online environment [8].

Moreover, a study conducted on Romanian students [8], also revealed that even though students frequently used E-learning platforms, in certain circumstances they would have preferred to use other online platforms in order to compensate for the disadvantages of the E-learning platform, such as technical problems, lack of proper functions that allow collaborative learning. In this context, it can be inferred that social networks like Facebook and Instagram could be used as complementary tools in the educational process.

Taking into account the aspects previously described, we considered relevant and necessary a research on the way social networks could be integrated in the online teaching-learning process, from the perspective of students. In this regard, the aim of the paper was to identify the attitude of Romanian students towards the use of Facebook and Instagram in the educational process and the circumstances in which students would prefer for this platforms to be used by them and their teachers. Facebook and Instagram were the focus of our research because they are two of the most popular and used social networks worldwide [10].

Thus, more specifically, having the above mentioned purpose, the paper focuses on the way Facebook and Instagram can be integrated in the process of teaching and learning in Romanian universities, and on identifying aspects related to how students would like to use these tools, what they would like their teachers to post on these platforms, and what types of activities would they prefer to carry out on them.

## Clarifying the concept of social networks

Social networks are widely used today by a significant number of people, their usage purpose diversifying over time. The term social network was utilized for the first time by the anthropologist J.A. Barnes, who was more interested in studying the informal relationships between people [11]. Social networks have been defined by Boyd and Ellison, as web based services that offer people the opportunity to build a public or semi-public profile in a limited system, to have access to a list of other users they can communicate with and to view and scroll through their list of connections and those made by other users [12]. Social networks are also platforms

that provide a social community for those interested in a particular topic or for those with common interests [13]. Liccardi defines social networks while referring to the concept of trust, considering that these types of sites are social structures made of nodes represented by individuals and that they are based on the trust between the members and the strength of the relationship [14]. Social networks can also be seen as sets of digital representations of individuals, referring to the registered users who are connected by taking into account the data about their activities, preferences or common communication [15].

The first social network dates back to 1997 and it is linked to the name of Andrew Weinreich, who created the site Sixdegrees.com, name related to the concept of six degrees of separation [16]. The concept originates from the experiment of psychologist Stanley Milgram, and it states that two people randomly chosen are connected to each other by a string of about six people [17]. The site allowed users to create a profile, list friends and even if it only functioned for three years, it was the first site that met the definition of a social network [18–20]. Ever since, social networks sites have gained popularity, the most popular being Facebook, closely followed by Instagram [10].

Often, the concept of social network is used as a synonym for the concept of social media, although there are some differences between them. Social media can be considered the content that people upload, can be viewed as a one-to-many communication, while social networking focuses on engagement, on creating relationships [21]. Social networking sites create virtual connections between people who may or may not know each other, individuals mainly using them for socialization, entertainment, updates regarding their contacts or finding information [22]. Social media is used, in general, to describe collaborative media creation and sharing on a large scale that includes social networks sites as well as other types of media as news blogs [23]. Social media can be seen as a platform for broadcasting information, while social networks are platforms where communication has a two way nature [24].

## Social networks as tools in educational process

Initially designed for the purpose of socializing, social networks seem to move to other fields, including education [25]. Generally, most universities today use information and communication technology, including social networks, in the process of teaching and learning in order to improve the learning process by making courses more accessible [26]. For instance, the annual report on online learning in higher education showed that in 2016, two thirds, 67% of students in the United States took at least one online course [27]. Even more, a study on the use of social networks developed at a university in Croatia, showed that 42. 17% of the respondents appreciated the possibility of using social networks for educational purposes as being a good possibility [28]. Thus, the academic experiences of students can be influenced by online communication, with social networks helping them accommodate easier to university life, and by creating connections between students, their peers and teachers [4]. Apart from being used for knowledge transfer, social networks are also used for more collaborative and higher learning as well as in the process of developing cognitive skills [5].

In comparison with the conventional educational system, social networks have a greater impact on both teachers and students as they offer the possibility to connect and collaborate in an easier way [29]. Students no longer use social networks only for connecting and interacting with friends or relatives [30], and these sites are nowadays also adopted by teachers because teaching and learning without technology often seems uninteresting [31]. Thus, a challenge that teachers must face is finding techniques to increase students' interest [29]. Another challenge of teachers is to transfer and use in the online environment, the methods utilized in the traditional teaching process [32].

Even more, with the development of online courses, it is important to understand the methods students use to learn and the manner in which they think in an environment where learning is very different from how it can usually be done in traditional courses [33]. Teachers have to take account of the fact that students have different strategies and abilities of learning, and negative experiences of them or their colleagues, whether coming from technical issues or difficulties in solving the assignments, can discourage them from learning online or having a positive attitude towards this type of education [34]. When teaching online, teachers must provide clear instructions, the content of the course must be delivered in a smooth, simple manner [35], and adapted to focus on the benefits offered by the environment [36]. Special attention should be given to the language used in teacher-student communication, this should be clear, without ambiguities [37], and using scientifically acknowledged terms for the specific field of teaching [38].

Besides being used for dissemination of information [39], social networks can improve students' level of engagement during courses, can enhance the educational process by allowing teachers to share with students varied course materials, and they can also improve collaboration between students, by allowing them to chat in real time that facilitates effective feedback and comment on the posts of their peers [6]. Social networks help teachers and students maintain continuous contact while setting up new spaces for learning; [40] they make learning interactive [41], and through them students can access and read the content taught at any date and time [7]. However, social networks can also have a negative impact while being used as educational tools. Studies [42, 43] on the use of social networks and their impact on student's grades showed that they negatively affect student's academic performance. Those using platforms such as Facebook registering lower grade-point averages, while other studies [44, 45], found no connection or significant relationship between the amount of time spent on social networks and grades, thus providing contrasting results on this matter.

## Facebook and Instagram as educational tools

It is a fact that Facebook is the most popular and largest social network nowadays, reaching a wide number of people who use it for multiple purposes. According to a study on the global internet use, in January 2020 there were 3.8 billion social media users and Facebook, having 2.44 billion monthly users. It remains the social network most used worldwide [10].

Facebook was originally created within Harvard University's grounds, [46] restricted only to Harvard students, and allowing them to create personal profiles, interact, socialize, and contact other students [47]. In educational institutions Facebook has been used with the intention to extend communication lines with students [48]. From a student's perspective, teachers could use Facebook for creating groups where presentations related to the lectures could be posted, as well as provide information about different changes with respect to the development of the courses [49]. Moreover, teachers should take advantage of the social potential of Facebook, [50] that provides the opportunity to build stronger connections between them and the university, thus students are more willing to study and complete their degree at their current university [51]. Despite its social potential, Facebook can have an educational potential by fostering the development of collaborative learning models, by offering a space where learning and teaching can be done in a more pleasant and comfortable way [52].

Facebook usage generates habits [53], and the platform is used by numerous educational institutions as a mean for students to acquire knowledge [54]. Currently there is a high interest for studying the effects of Facebook and what students use it for [55]. According to a study on Facebook as an academic tool [56], the main users of the network are between 18 and 24 years, and considering the findings of the research a starting point on this matter, the researchers

showed that some students believe that Facebook is a much more proper medium for social learning than the real one; they preferring to receive tasks on this platform and, because they prefer this medium for discovering information, accomplishing school tasks through this platform. Another study supports the idea that students also use Facebook for discussing educational content, [57], while another study reveals that students view Facebook rather as a tool that can help them acclimate to university life [58]. Even more, a large number of students access Facebook in order to find out more information about people they had recently met and to stay in touch with old and present friends [59]; additional studies [60] show that students would rather interact with friends on public wall spaces, with this type of communication being twice as frequent as exchanging information through private messages, over personal or face-to-face interaction. In addition, it is thought that Facebook is the social network best suited for students [61]. This statement is sustained by a study which included students using a Facebook group for academic purposes confirmed an improved student interaction with their teachers, and, thus, concluding that the use of social networks could refine their learning experience [62].

Related to Facebook use, six categories were found to be important, among them being: experience, social learning, initiating relationships or learning about others, [63] while the educational use of this platform has notable positive connection with its use for interaction and exchange of information and resources [64]. Recognizing the educational potential of Facebook, teachers must find ways to efficiently use it and understand how to interact with students, so they create positive attitudes towards this type of learning [65]. However using only Facebook as a teaching tool may not be as effective as combining it with others social networks.

While Facebook is still the most popular network in the world, it is known that younger audiences also prefer social networks which are more centered on visual content. Thus, there has also been a notable increase in the number of students who prefer Instagram as an educational tool [66]. Instagram is a photo-video sharing platform launched in 2010, currently owned by Facebook, that allows users to post visual content, 24 hours stories, and privately communicate with their friends [67]. The platform registering in 2020 over 1 billion monthly active users [10]. Instagram is especially designed for mobile use; it is a platform more orientated to a mobile experience. One of the reasons for its rapid ascension is the development and widespread use of smartphones [68]. However, Instagram is about more than just photo sharing. The platform offers the possibility to create communities in many domains, and the ability for users to broaden their contact list, exchange information, share knowledge through public or private messages, live videos [69]. Additionally, through its emphasis on visual content, and through the actions of the users (taking the photo, editing a photo, and writing a description) Instagram can help improve student's spatial or linguistic intelligence [70]. While students today are considered digital natives who heavily use social networks, [71], professors should integrate such networks as teaching tools in order to enhance their learning experience and give students the chance to share ideas and opinions through a medium that they consider pleasant and interesting [72].

Due to its features, Instagram can be used as an educational tools that can encourage the development of new methods of learning and teaching, making these processes unique and attractive [73]. A previous study on the efficiency of Instagram in the field of education indicated that including this platform in the teaching process helped improve students writing, creativity, and story development by approaching diverse topics [74]. Other studies also support the idea that Instagram can have positive effects on students, showing that this tool can help them socially adjust [75] and can assist in the process of learning new languages [76]. However, from the student's perspective, Instagram is more useful in the educational process

if it is used as an auxiliary tool for traditional courses [77]; when asked in a study [78] to rank social media apps, teachers and students mentioned Instagram as being the second social network that they use most, after WhatsApp and Facebook. Thus, according to previous studies, Instagram is used in education as a tool for improving student's ability to write, to learn a new languages, and aid in research development.

## Materials and methods

### Sampling and data collection procedures

The present cross-sectional study was conducted on students of Romanian public higher education institutions. The questionnaire was administered online (as a link in Google forms) and disseminated through Facebook (pages of the universities), during the second semester of the 2019–2020 academic year. Participants of the study received information at the beginning of the questionnaire about the purpose of the survey and the informed consent. The average time needed to answer the questionnaire was 30 min. The research received the approval of the Ethics Commission in social research from Transilvania University of Brasov, Romania.

The questionnaire included a convenience sample made up of 872 students. 165 (18.9%) students were male, and 707 female (81.1%).A total of 708 (81.2%) students are in the Bachelor Primary Education program, 135 (15.5%) are enrolled in a Master program, and 29 (3.3%) are in a PhD program. The selected 847 (97.1%) students have a Facebook account, and 86.7% use daily their Facebook account. Also, 760 (87.2%) students have an Instagram account, and 84.7% of them use daily their 13 Instagram account. This information, together with other characteristics of the participants are presented in Table 1.

### The research instrument

Based on the data gleaned and a series of mechanisms previously used by researchers in order to analyze students' attitude towards the use of Facebook and Instagram in general, [79–82],

**Table 1. Description of sample characteristics (n = 872).**

|  | Category | Count | Percentage |
|---|---|---|---|
| Gender | Female | 707 | 81.1% |
|  | Male | 165 | 18.9% |
| Degree | Bachelor | 708 | 81.2% |
|  | Master | 135 | 15.5% |
|  | PhD | 29 | 3.3% |
| Age | 18–22 years | 621 | 71.2% |
|  | 23–25 years | 103 | 11.8% |
|  | over 25 years | 148 | 17% |
| Field of study | Social and Human sciences | 790 | 90.6% |
|  | Other sciences | 80 | 9.2% |
|  | na | 2 | 0.2% |
| Facebook account | Yes | 847 | 97.1% |
|  | No | 25 | 2.9% |
| Use of Facebook account | Daily | 734 | 86.7%* |
| Instagram account | Yes | 760 | 87.2% |
|  | No | 112 | 12.8% |
| Use of Instagram account | Daily | 646 | 84.7%* |

* Percentages are calculated from the total of those who stated that they have an account.

we developed a questionnaire that measured the attitude of students towards the use of the these two platforms in the educational process.The questionnaire can be found in the "S4 Appendix. Questionnaire English version" and "S5 Appendix. Questionnaire Romanian version". The questionnaire is comprised of four scales, two for each of the social networks analyzes, scales that focused on the use of Facebook and Instagram in the online teaching and learning process by teachers and students. The scales and their items were pilot tested among 50 students who had an active Facebook and Instagram account. All students understood the questions, and they did not find the questions difficult to answer. However, based on their feedback, we reformulated questions by inserted or deleted certain elements. Thus, for each of the two social networks included in the study, a set of items referring to different contexts in which the platforms can be used were established. All measures employ a seven-point Likert scale, ranging from 1 ("disagree very strongly") to 7 ("agree very strongly"), and each respondent had to evaluate, express their level of agreement with respect to questions which described a specific context in which Facebook and Instagram could be used for academic purposes ("Please indicate how much you agree or disagree with the following statements").The last part of the questionnaire was comprised of a series of socio-demographic variables: gender, degree level, field of study, and the frequency of using Facebook and Instagram platforms. This information was used in order to identify differences of attitude among certain groups.

## Statistical analyses

The data was analyzed using IBM SPSS Statistics 23. The Exploratory Factor Analysis (EFA) indicated that the data for each of the four mechanisms were suitable for two factor analysis. Bartlett's tests of sphericity and the associated significance probabilities (p = 0.000) indicated that the observed correlation analyses were statistically significant. Kaiser-Meyer-Olkin's values was greater than 0.90. Varimax rotation was performed. Principal components analysis was performed and produced a two-factor solution with eigenvalues greater than 1. All items with saturation higher than 0.6 were retained for interpretation. Some indicators were removed (f20.p, f21.p; f23.p, f24.p, f19.s, f21.s; i17.p, i18.p, i17.s) and we had two reasons for removing them. One of the reasons is related to the correlation of the coefficients of items with the factors, which are nearby 0.60 or lower. Another reason is related to the fact that these indicators are correlated with both factors extracted (the correlation coefficients are approximately equal). Looking more closely at these items we see that they are formulated generally, and because of these, students may have different representations to which to refer. Therefore, initial versions of the scales are presented in "S1 Appendix. Initial versions of the scales", and the final versions of scales are presented in "S2 Appendix. Exploratory factor analyses item loadings, Reliability, and Explained Variances.". For each scale the factors were labeled: *The use of social networks during the didactic activity*, *The use of social networks in extracurricular activities and career development*, to name a few (see Table 2). For each of the fours scales internal reliability was tested using Cronbach's alpha. The factor loadings, the internal-consistency test result (Cronbach's alpha), the means and standard deviations for all items are shown in "S2 Appendix. Exploratory factor analyses item loadings, Reliability, and Explained Variances".

For all factors corresponding on each scale a mean index was created [83], ranging from 1 to 7, representing students' attitude towards the use of Facebook and Instagram by teachers and students in the context of carrying out didactic activities and extracurricular activities exclusively online. The descriptive statistics (mean and standard deviation) are presented in Table 2.

Comparisons between mean index values are made with the t test and results are presented in "S3 Appendix". Comparisons were made also depending on gender, field of study, level of

**Table 2. Descriptive statistics-aggregate data.**

| | Name of aggregate variables | N | Mean | S.D. |
|---|---|---|---|---|
| | *7-strong positive attitude* | | | |
| | *1-strong negative attitude* | | | |
| Teachers' use of Facebook | AD_Fp (Didactic activity) | 872 | 5.03 | 1.87 |
| | EC_Fp (Extracurricular information and career development) | 872 | 5.52 | 1.59 |
| Students' use of Facebook | AD_Fs (Didactic activity) | 872 | 4.81 | 1.91 |
| | EC_Fs (Extracurricular information and career development) | 872 | 5.53 | 1.67 |
| Teachers' use of Instagram | AD_Ip (Didactic activity) | 872 | 4.25 | 2.15 |
| | EC_Ip (Extracurricular information and career development) | 872 | 4.7 | 2.07 |
| Students' use of Instagram | AD_Is (Didactic activity) | 872 | 4.00 | 2.18 |
| | EC_Is (Extracurricular information and career development) | 872 | 4.63 | 2.10 |

degree (Bachelor/Master+PhD), and frequency of using the Facebook and Instagram accounts. Thus, only the results for the variables where significant differences where statistically identified with the t-test are presented (see Tables 3 and 4).

We considered it appropriate to compare respondents according to their gender, because, generally, in many countries (as well as in Romania), women are more active users of social networks [10]. Thus, we thought that this fact may also have an impact on women's' attitude towards using Facebook and Instagram as educational tools. However, the results of our study indicate no statistically significant difference between male and female students.

Comparisons were also made according to the respondents' field of study because certain courses belonging to the social or humanistic domain might be better suited for the type of interaction presumed by social networks, rather than courses from the technical field. Thus, results of our study indicate no statistically significant difference depending on this variable.

Even more, we considered important a comparison depending on the level of degree of the respondents (Bachelor/ Master +PhD), because the profiles of the two categories of students

**Table 3. Significant t-test results: Facebook/Instagram scale–degree level.**

| | | | | | | | | t-test for Equality of Means | | | |
|---|---|---|---|---|---|---|---|---|---|---|---|
| | Group | N | Mean | S. D. | t | do | p | Mean Difference | Std. Error Difference | CI4 | |
| | | | | | | | | | | Lower | Upper |
| AD_Fp | Bachelor | 708 | 4.97 | 1.86 | -2.07 | 870 | .03 | -.33 | .16 | -.65 | -.01 |
| | Master+ PhD | 164 | 5.31 | 1.89 | | | | | | | |
| EC_Fp | Bachelor | 708 | 5.47 | 1.6 | -1.83 | 870 | .06 | -.25 | .13 | -.52 | .01 |
| | Master+ PhD | 164 | 5.72 | 1.54 | | | | | | | |
| AD_Fs | Bachelor | 708 | 4.73 | 1.91 | -2.53 | 870 | .01 | -.41 | .16 | -.74 | -.09 |
| | Master+ PhD | 164 | 5.15 | 1.87 | | | | | | | |
| EC_Fs | Bachelor | 708 | 5.46 | 1.69 | -2.53 | 256 | .01 | -.35 | .13 | -.62 | -.07 |
| | Master+ PhD | 164 | 5.81 | 1.58 | | | | | | | |
| AD_Ip | Bachelor | 708 | 4.22 | 2.15 | -.90 | 870 | .36 | -.16 | .18 | -.53 | .19 |
| | Master+ PhD | 164 | 4.39 | 2.19 | | | | | | | |
| EC_Ip | Bachelor | 708 | 4.66 | 2.06 | -1.12 | 870 | .26 | -.20 | .18 | -.55 | .15 |
| | Master+ PhD | 164 | 4.86 | 2.11 | | | | | | | |
| AD_Is | Bachelor | 708 | 3.99 | 2.15 | -.72 | 234 | .46 | -.14 | .19 | -.53 | .24 |
| | Master+ PhD | 164 | 4.12 | 2.30 | | | | | | | |
| EC_Is | Bachelor | 708 | 4.58 | 2.08 | -1.49 | 870 | .13 | -.27 | .18 | -.63 | .08 |
| | Master+ PhD | 164 | 4.85 | 2.17 | | | | | | | |

**Table 4. Significant t-test results: Facebook/Instagram scale–daily use/rarely use.**

| | Group | N | Mean | S. D. | t-test for Equality of Means | | | | | | |
| | | | | | t | df | p | Mean Difference | Std. Error Difference | CI4 | |
| | | | | | | | | | | Lower | Upper |
| AD_Fp | Daily | 734 | 5.23 | 1.75 | 4.52 | 140 | .00 | .90 | .19 | .50 | 1.29 |
| | Rarely | 113 | 4.33 | 1.98 | | | | | | | |
| EC_Fp | Daily | 734 | 5.67 | 1.48 | 4.07 | 845 | .00 | .61 | .15 | .32 | .91 |
| | Rarely | 113 | 5.05 | 1.66 | | | | | | | |
| AD_Fs | Daily | 734 | 4.98 | 1.82 | 3.77 | 845 | .00 | .70 | .18 | .33 | 1.07 |
| | Rarely | 113 | 4.28 | 1.99 | | | | | | | |
| EC_Fs | Daily | 734 | 5.66 | 1.58 | 2.43 | 141 | .01 | .42 | .17 | .07 | .77 |
| | Rarely | 113 | 5.23 | 1.75 | | | | | | | |
| AD_Ip | Daily | 734 | 4.47 | 2.11 | 2.18 | 761 | .02 | .46 | .21 | .04 | .88 |
| | Rarely | 113 | 4.01 | 2.12 | | | | | | | |
| EC_Ip | Daily | 734 | 4.85 | 2.00 | .92 | 761 | .35 | .18 | .20 | .-.21 | .58 |
| | Rarely | 113 | 4.66 | 2.09 | | | | | | | |
| AD_Is | Daily | 734 | 4.19 | 2.16 | 1.94 | 761 | .05 | .42 | .21 | .00 | .84 |
| | Rarely | 113 | 3.77 | 2.11 | | | | | | | |
| EC_Is | Daily | 734 | 4.76 | 2.03 | .42 | 761 | .67 | .08 | .20 | -.31 | .49 |
| | Rarely | 113 | 4.67 | 2.12 | | | | | | | |

vary. Master and PhD students are mainly students who are already employed, students who have reached a more mature age, to whom it may be easier and more convenient to use social networks in a formal, educational context. A previous study conducted in Romania also pointed out that these students prefer using online platforms in the educational process, more than students who are enrolled in educational programs with other levels of degree [8]. Results of our study indicate statistically significant differences depending on this variable (see Table 3).

Differences in students' attitudes were also tested according to the frequency of the use of Facebook and Instagram accounts. Thus, two other variables were created with two categories (daily and less often), and those who did not use Facebook or Instagram were eliminated from our analysis. In this regard, we expected students who use Facebook and Instagram daily, to be more open and willing to use these social networks in the educational field, because they are more familiar with these platforms. Results of our study show statistically significant differences depending on these variables on some items (see Table 4).

## Results

### What is the attitude of students towards the use of Facebook and Instagram in the educational process?

The means (Table 2) indicate that most students tended to have slightly positive attitudes towards Facebook, since the mean scores all greater then 4 (on a scale from 1–7), ranging from 4.81 to 5.53, but students did not provide high scores, or scores close to the maximum score possible-7, to none of the dimensions analyzed. Thus, this results may suggest that students believe that the use of Facebook could be beneficial during the educational process, but most likely students consider it a complementary tool that could be used occasionally.

The results of the research revealed that Facebook is suitable for developing extracurricular activities and activities for career development, regardless of whether it is used by teachers (M = 5.52, SD = 1.59) or students (M = 5.53, SD = 1.67), and that it is considered less suitable

for the didactic activity, for the formal educational process (M = 5.03, SD = 1.87; M = 4.81, SD = 1.91; t(871) = -12.39, p = .00, t(871) = -15.76, p = .00). Even more, when referring to the didactic activity, students consider more appropriate the use of Facebook by teachers (M = 5.03, SD = 1.87), for posting diversified information and for involving students in varied activities, than the use of Facebook by students for carrying out course related activities with their peers (M = 4.81, SD = 1.91; t (871) = 5.66, p = .00) (S3 Appendix). In the case of using Instagram as an educational tool, students have a more reluctant attitude, ascribing lower scores to the affirmations included in the survey in this regard. The means indicate that most students tended to have less positive attitudes towards Instagram as an educational tool, the mean of the scores being around 4 (on a scale from 1 to 7), ranging from 4 to 4.7 (see Table 2). Hence, even though today's generation of students are considered to be digital natives and heavy users of Instagram, students do not think of this social network as being an appropriate tool for carrying out the educational process. Moreover, when referring to Instagram, students have more distinct and divergent opinions, the standard deviation having higher values than in the case of the student's responses regarding Facebook.

Similar to their responses for using Facebook as a tool in the teaching and learning process, students are of the opinion that Instagram is a better suited platform for developing extracurricular and career development activities, regardless of whether Instagram it is used by teachers (M = 4.7, SD = 2.07) or students (M = 4.63, SD = 2.10). Students believe Instagram is unsuitable for unfolding the formal educational process (M = 4.25, SD = 2.15; M = 4.00 SD = 2.18; t(871) = -12.23, p = .00, t(871) = -14.57, p = .00). Furthermore, similarly to the use of Facebook for academic purposes, taking into account the didactic activity, students believe that Instagram is more appropriate to be used by teachers (M = 4.25, SD = 2.15), for posting information and engaging students in diverse activities, than to be used by students for developing activities together with their fellow colleagues, or other types of activates (M = 4.00, SD = 2.18; t (871) = 6.53, p = .00) (S3 Appendix).

The results of the research did not reveal any differences depending on the gender of the respondents or their field of study, but according to the level of degree and frequency of using their Facebook and Instagram accounts, opinion differences were revealed (see Tables 3 and 4). In this regard, it can be inferred that master's or Ph.D. students are much more open to the use of Facebook in the educational process by both teachers (M = 5.31, SD = 1.89), and students (M = 5.15, SD = 1.87; M = 5.81, SD = 1.58), than Bachelor students. (M = 4.97, SD = 1.86; M = 4.73, SD = 1.91; M = 5.46, SD = 1.69; t (870) = -2.07, p = .03; t (870) = -1.83, p = .01; t (256) = -2.53, p = .01). However, when it comes to Instagram, students' opinions no longer differ depending on these variables.

Taking into account the use of Facebook according the frequency of use, the results suggest, as we expected, that those students who use their Facebook account daily have a more open attitude towards utilizing this tool in the educational process, regardless of the dimension analyzed (p < .05). Given the use of Instagram, students who access daily their account have a more positive attitude towards its use in the didactic activity by teachers (p<05). Thus, apart from the differences previously mentioned, no other statistically significant differences were found between the two categories (see Tables 3 and 4).

## What are the contexts that students consider appropriate for the use of Facebook and Instagram in the educational process?

After conducting the factorial analysis, and after we analyzed the items corresponding to each factor of the scale (S2 Appendix. Exploratory factor analyses item loadings, Reliability, and Explained Variances), we identified certain contexts, circumstances related to the educational

process in which students believe Facebook and Instagram can be used for educational purposes. These contexts are presented in the following sections.

**The use of Facebook and Instagram by teachers.** In consideration to the use of social media for formalized educational purposes, students believe teachers could use Facebook for providing answers to their questions concerning class projects and tasks, inviting professionals in their respective field for debates on varied subjects, sharing their experience related to professional activity, conducting surveys/polls on certain matters that concern the development of the courses and seminars, and, finally, informing students about any type of changes regarding the courses or the deadlines for their projects.As we previously stated, our research revealed that Instagram is not given a central place in the educational process, but this platform could be used in certain circumstances, occasionally, in order to diversify the activities that are carried out during the courses or the seminars. If it were to use Instagram during the didactic activity, during the formal educational process, students believe that teachers could make use of the platform by inviting field specialists to post photos, diverse presentations or videos related to the theme of the course, by sharing personal experiences related to their professional activity, by posting photos, presentations or videos made during specific curricular and extra-curricular projects developed together with the students, but also by offering students information about organizational and administrative aspects of the faculty.

A context in which students believe that both Facebook and Instagram could be used by teachers is represented by extracurricular activities. More specifically, these social networks could be used by teachers in order to post announcements about jobs of interest to students, about internships opportunities, personal development workshops, volunteering opportunities, about various projects of interest at community level and partnerships that the faculty has. Even more, students consider that on these platforms teachers could also post information in order to promote socio-cultural events, or other extracurricular events that are carried out or that will be carried out by their faculty.

**The use of Facebook and Instagram by students.** In consideration to the use of Facebook and Instagram by students as a didactic activity, students consider these social networks to be an appropriate place for them to manage interaction and group collaboration.Thus, students would like to use Facebook and Instagram in order to chat with their peers about the projects and tasks that they must accomplish, to suggest diverse debate topics related to the course curriculum, to share ideas regarding the didactic activity and the way the process unfolds, to carry out surveys on certain course and seminar themes, to post announcements about course and deadlines related changes.

In the context of using Facebook and Instagram for developing extracurricular activities, students believe these platforms are suitable for them to post announcements about jobs of interest to other students, about internship opportunities, personal development workshops, volunteering opportunities, information about projects of interest at the level of the entire community. Furthermore, students think that they could use the above mentioned social networks in order to post information with the purpose of promoting socio-cultural events organized by their faculty, to post information about other extracurricular present and future events carried out by their faculty, but also to interact with their colleagues or other students. Thus, considering the dimensions of extracurricular activities, our research revealed that students would like for Facebook and Instagram to be used by their teachers for conducting activities that are similar to the ones students would like to conduct on these platforms.

## Discussion

Previous studies on social networks use in higher education revealed that many universities use and integrate Facebook into the educational process [54] and that students have positive

attitudes towards using Facebook during the educational process [49, 56, 62]. Our research revealed that Romanian students would rather use Facebook as a complementary tool for the teaching and learning process, a tool that could be mainly used for posting information or conducting extracurricular activities and less used during the didactic activity.

Hence, if other countries opt for the integration and use of social networks for educational purposes [25, 54, 55, 78], the use of such platforms in higher education institutions in Romania is rather modest, and often resumes to posting administrative information. Thus, one of the reasons for conducting the present study is represented by the desire to carry out a research that could complement and act as a response to a similar study [56], which also focused on students' motives for using Facebook. In other words, our desire was to conduct a research that could support a previous study which also addressed the subject of the use of Facebook by students. The previous study [56] addressed the matter from the perspective of students, and focused on the question of what students should post on their Facebook accounts in order to maintain a proper interaction with their teachers. Even more, in the study were involved some of the researchers involved in the present study, and the purpose of the study was to identify how students perceive the use of Facebook for academic purposes. The results of the study revealed that more than half of the respondents (57. 3%) were open to receiving tasks and projects through messages sent through Facebook, and that 30% of the respondents considered that Facebook can represent a comfortable environment that could also increase their motivation to research, discover or complete school projects. In this regard, our study also approaches the matter of using social networks for academic purposes from the perspective of students, but it focuses on the question of what should teachers post on platforms such as Facebook and Instagram in order to interact with their students. Hence, our research was conducted by taking into account the perspective of students regarding the ways they would like Facebook and Instagram to be used as educational tools in general, and in the context of the pandemic in particular. These two platforms manage to register impressive numbers of users worldwide [10], and they are the subject of our research because they provide users with a wide range of functions and options. Further, we focused solely on Facebook and Instagram and not on other platforms such as YouTutbe, Tik Tok, Steemit [84], Researchgate, LinkedIN, Skillshare or Coursera, and we further present several reasons for this. Hence, YouTube was not included in our research because, even though, through the functions and instruments that it incorporates, it can provide users with diversified content. As a consequence, we considered that this platform is less interactive because it does not allow the creation of groups, and thus it could hinder the process of real time communication between students. In regard to Tik Tok, while we considered the educational benefits of the platform, we decided to exclude it from the study because it is a relatively new platform and people's response to the application is uneven. Even more, we believe that compared to Instagram and Facebook, Tik Tok has this novelty character that can lead to measurement errors. This errors might appear because people are generally excited about everything that is new in matter of technology, and thus their answers might have been influenced by the novelty of the platform Tik Tok. Next, as for Steemit and Skillshare, they were not included in our research because they are used to a very little extent in Romania. While Researchgate is a beneficial platform for promoting research and spreading knowledge trough the scientific community, it is not a widely known application with diverse functions. LinkedIn is a more formal, business platform, that could be used by students and teachers for searching for jobs and internships; however, our interest was in finding out how social networks more general and comprehensive, like Facebook, could be integrated into the educational process. Finally, Coursersa was not a focus of this particular study because it centralized its format specifically on educational purposes, rather than social network outreach.

Taking into account the aforementioned aspects, we argue that, compared to other social networks, Facebook and Instagram can better support the learning process. These platforms incorporate a wide variety of functions that allow the creation of diversified content, they allow students to properly communicate and collaborate with each other during courses. For example, on Facebook, teachers could create groups and students could work in teams in order to fulfil certain tasks, that can later be presented to the other teams on a live meeting. Last but not least, students are familiar with these social networks. Even though they usually use them to relax and communicate with their friends, because they are already accustomed to them, students may be more relaxed, more creative and more willing to learn in a familiar environment.

In a previously mentioned study [56], conducted in Romania in 2011, is showed that 70% of the respondents were of the opinion that Facebook was a low-cost tool that could be used to promote knowledge in higher education, but only 26.7% of them considered Facebook an appropriate instrument of change in higher education, an instrument through which changes in strategies, and attitudes during the educational process could be achieved. However, we believe Facebook can be more effectively used for academic purposes. The findings of our research revealed a slightly positive attitude of students towards the use of this platform in education, even if more as a complementary tool, as well as certain contexts in which students would like Facebook to be used by them and their teachers.

Nonetheless, in the context of the shift to exclusively online learning, studies conducted in Romania [8, 9] revealed that students are more interested in learning through more formal means and thus they would rather prefer to use platforms dedicated to online learning. Still, due to the amount of technical issues that they encounter when using such platforms, students would also like to use social media platforms. Thus, even if E-learning platforms can facilitate the educational process, they still have some options that sometimes can hinder the educational process because of their poorly way of functioning, such as the videoconference option when large numbers of students are connected. In this situations, students would also be willing to use social networks such as Facebook or Instagram.

According to our findings, students prefer using Facebook exclusively for educational purposes, opting to exclude the use of Instagram. Because students prefer Facebook, and because it is better suited for students academically [61], this platform may aid instructors in student socialization and interaction. Our study also revealed acceptance of these social platforms by master's and Ph.D. students, which concurs with previous studies conducted in Romania [8], which supported the use of online platforms in students with advanced educational degrees as opposed to Bachelor's students."

With respect to the use of Instagram as an educational tool, students expressed reservations towards this social network platform, considering its usefulness in the educational process occasionally, on specific activities that are usually extracurricular activities. Based on this inference, we found, like previous studies related to Instagram use for educational purposes, [77], that Instagram too can be considered a complementary tool for the educational process. In the case of Instagram, it was found that students' study degree does not have an influence on their attitude towards using this platform in the educational process. Even more, students who use their Facebook and Instagram accounts daily are of the opinion that these platforms could also be used for academic purposes. Students' attitudes towards the use of social networks for educational purposes relates to their usage and familiarity with their programs.

Furthermore, when addressing the subject of the use of social networks, regardless of the purpose for which they are being used, a discussion about students' addiction to platforms such as Facebook and Instagram is relevant and necessary. Hence, a previous study which took into account the mathematical model of people's Dopamine System model, analysed the

behavior of Facebook users belonging to 18 Facebook groups from domains such as education, sport, entertainment, work or news, and revealed that the notifications people receive on social networks can generate addiction [85]. With respect to the academic field, previous studies also showed that university students can become addicted to Facebook, according to some of their personality and psychological traits such as depressive character, and that the more time students spend of Facebook, the more likely are to develop Facebook addiction [86]. Social network addiction is a serious problem, and while taking into account the findings of previous studies, we argue that, by using social networks in the teaching-learning process, teachers could actually help students overcome this problem. Hence, it is known that students spend many hours on platforms like Facebook, study [86] showing that the majority of the respondents daily used Facebook for more than 4.5 hours. Thus, students consume content posted on social network for many hours and can slowly become addicted by saying to themselves that they will only watch one more post. In this regard, students use social networks for varied purposes, but we are of the opinion that the use of these platforms for educational purposes should not increase students' level of addiction. On the contrary, because they are platforms where the content is not restricted, or it is restricted to a very little extent, teachers can share diversified course related content, but also other type of information that could help students understand the problem of addiction, that could help them cope with it and teachers may encourage students to use social networks like Facebook and Instagram in order to expand their knowledge, to share ideas, and not just use them for entertainment purposes. In other words, universities and teachers should focus more on findings ways through which students could be encouraged to use social networks in more positive and beneficial ways, and thus, by making them use Facebook and Instagram for academic purposes, professors could also teach their student about addiction, they could offer them advices and counselling.

Taking into consideration the aspects mentioned above, certain issues can be addressed. The first issue refers to whether Moodle platforms provide better support than social networks for carrying out the educational process. Hence, we argue that, Moodle platforms are suitable for conducting the educational process and the aim is not to eliminate the use of Moodle platforms for delivering courses, but to add value to the educational process by supplementing it with the use of social networks. Because of their nature, and because students are familiar with these platforms and enjoy using them, social networks could enhance the learning process, and in this regard, they could be used as complementary tools. Even more, another issue related to the use of Moodle platforms in the educational process is represented by the fact that not all educators have the necessary skills in order to properly use the platforms. However, in order to help teachers understand how to use such platforms and to better collaborate with students through them, universities could organize certain training sessions for teachers, and they could offer teachers access to detailed guides that provide explanations on the way the tools integrated into the platform should be used.

Another issue that arises when approaching the subject of using Facebook and Instagram as educational tools, refers to the matter of whether students will be willing to use these platforms only because they would have a reasonable motive for using them, thus not developing a feeling of guilt for not studying. In this regard, as we previously mentioned, by integrating social networks in the educational process, we believe that students could be taught how to make better use of them. In this regard, the act of using social networks for academic purposes should not be seen as a method that helps students eliminate their guilt for not studying, because the activities that can be carried out through these platforms are activities meant to help students study. In other words, being encouraged and accepting to use social networks for educational purposes, does not mean that students will prefer to use such platforms because in this way they can eliminate the guilt for not studying, because the action of using Facebook and

Instagram for academic purposes itself is an action that the student made in order to learn and acquire knowledge.

## Conclusions

In the context of the pandemic, major changes were generated in the academic field and certainly, higher education institutions will continue to use online learning platforms in the future even if they will be allowed to return to the traditional way of conducting the educational process.

The return to the traditional way of learning will not exclude online learning platforms, but social networks such as Facebook and Instagram could also be used in some cases as complementary tools for the educational process. In this regard, the results of our research revealed that Facebook and Instagram could be integrated in the process of teaching and learning in Romanian universities, but only as auxiliary tools for the traditional process or for the use of E-learning platforms. The main context in which these social networks could be used are extracurricular activities, but they could also be used in contexts related the educational act in order to enhance the process of learning, to increase students' level of creativity or to develop some of their abilities, such as writing skills.

The quality of the educational process in the online environment depends on multiple factors, such as technical conditions or the strategies used by teachers to capture students' attention. By understanding the way students relate to the integration of Facebook and Instagram in the process of teaching and learning, teachers could adapt their methods of delivering the courses and seminars so as to develop new ways of sparking students' interest, and to improve the educational act. However, the parallel use of multiple platforms may increase students' level of stress. In order not to damage the educational process, it is of utmost importance for teachers to identify the preferences, and willingness of each groups of students to use social networks as complementary learning tools, so as to increase the performance of students and their interest towards the subject of the courses.

Turning to the theoretical and practical implications of our paper, this research can contribute to the literature by providing information on the use of social networks in the teaching-learning process of higher education institutions in general, and in the context of the pandemic in particular. With reference to the practical implications, by examining students' preferences regarding the way they would prefer Facebook and Instagram to be used by them and by their teachers for academic purposes, our paper provides relevant information that may help teachers better understand how use social networks in order to enhance the educational process and increase students' interest.

### Limitations and future research directions

The study we conducted also has some limitations that may represent subjects for future researches. Because our study is exploratory, the instruments used were in the first testing phase, and thus in the future the research design could be adapted in order to further validate the instruments. Also, other research methods could be applied so as to find interesting patterns in the data. For instance, focus-groups may bring more insights in order to understand the motivation behind the scores given to the items, and this information could be useful for improving the instruments used and for evaluating the psychometric properties. Even more, another limitation is given by the fact that the study was carried out only on Romanian universities, and thus a future research should take into account other European universities.

## Supporting information

**S1 Appendix. Initial versions of the scales.**
(DOCX)

**S2 Appendix. Exploratory factor analyses item loadings, reliability, and explained variances.**
(DOCX)

**S3 Appendix.**
(DOCX)

**S4 Appendix. Questionnaire English version.**
(DOCX)

**S5 Appendix. Questionnaire Romanian version.**
(DOCX)

## Acknowledgments

We kindly thank the respondents for participating in the study, especially because they agreed to voluntary participate in our study, without receiving any incentives.

## Author Contributions

**Conceptualization:** Claudiu Coman, Luiza Mesesan-Schmitz, Maria Cristina Bularca.

**Data curation:** Luiza Mesesan-Schmitz.

**Formal analysis:** Luiza Mesesan-Schmitz.

**Investigation:** Claudiu Coman, Maria Cristina Bularca.

**Methodology:** Luiza Mesesan-Schmitz.

**Project administration:** Claudiu Coman.

**Resources:** Maria Cristina Bularca.

**Supervision:** Laurentiu Gabriel Tiru, Gabriela Grosseck.

**Writing – original draft:** Luiza Mesesan-Schmitz, Maria Cristina Bularca.

**Writing – review & editing:** Claudiu Coman, Luiza Mesesan-Schmitz, Laurentiu Gabriel Tiru, Gabriela Grosseck, Maria Cristina Bularca.

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
