## [Decision Letter · Decision Letter 0]

11 May 2021

PONE-D-21-10280

Dear student what should I write on my wall? A case study on academic uses of Facebook and Instagram during the pandemic

PLOS ONE

Dear Dr. Coman,

Thank you for submitting your manuscript to PLOS ONE. After careful consideration, we feel that it has merit but does not fully meet PLOS ONE’s publication criteria as it currently stands. Therefore, we invite you to submit a revised version of the manuscript that addresses the points raised during the review process.

The manuscript needs a MAJOR REVISION. Please follow the suggestions given by the reviewers, in order to improve the readability of the paper.

We look forward to receiving your revised manuscript.

Kind regards,

Barbara Guidi

Academic Editor

PLOS ONE

Journal Requirements:

[We kindly thank the respondents for participating in the study and we thank Transilvania University of Brasov and West University of Timisoara for their support.]

 [The authors received no specific funding for this work.]

Reviewers' comments:

Reviewer's Responses to Questions

**Comments to the Author**

1. Is the manuscript technically sound, and do the data support the conclusions?

Reviewer #1: Yes

Reviewer #2: Partly

2. Has the statistical analysis been performed appropriately and rigorously? 

Reviewer #1: Yes

Reviewer #2: Yes

3. Have the authors made all data underlying the findings in their manuscript fully available?

Reviewer #1: Yes

Reviewer #2: Yes

4. Is the manuscript presented in an intelligible fashion and written in standard English?

Reviewer #1: No

Reviewer #2: Yes

5. Review Comments to the Author

Reviewer #1: Please change to (or something similar):

Line 61 Although E-learning platforms or social networks were used by higher education institutions in the educational process prior to the COVID-19 pandemic, due to the pandemic,

Line 66 Recent studies that focused on identifying the perception of Romanian students about online learning during the pandemic [8,9], revealed less positive attitudes towards this type of learning, students considering that the online educational process has less value than the traditional one, preferring instead to use E-learning platforms as a complementary method to traditional, face-to-face learning.

Please see attachment for additional comments

Reviewer #2: In this paper the authors propose a study concerning the usage of Facebook and Instagram in a university scenario. The paper is well written, however I have few concerns.

1) I understood that you focus the two most important social networking/media platforms, but why these two in particular? Why not using other platforms, such as YouTube, TikTok, Steemit, Researchgate, or Linkedin? Why not considering other platforms specifically tought for learning such as Skillshare and Coursera? In short, the paper should clearly state what Facebook and Instagram have to support the learning process that other platforms don't have.

2) In the Introduction the very first paragraph is not convincing (lines 50-60). In particular:

- Reference 3 states that for 70% of the students Facebook is not superior to Moodle, 90% of the students find Facebook not appropriate for learning (see the paper's table 2)

- Reference 6 discusses how social networking sites can be used in the learning process, but does not prove the usefulness.

I think the authors should replace these references with other more relevant.

3) Platforms such as Facebook and Instagram were shown to generate pethological addiction (see https://www.springerprofessional.de/en/discovering-the-impact-of-notifications-on-social-network-addict/18929896 ) also in the academic scenario (see https://www.sciencedirect.com/science/article/pii/S0736585314000021 ). How do you frame this problem in the usage of social networks for learning? Since this problem is mainly due to the "just one more post/picture/video" effect, where you keep consuming contents mindlessly for hours, what is the impact of using such an unrestricted platform for learning? Does Moodle and other such platforms not provide better support? How much of the interviewers were in favour of using, say, Facebook just because they can have access to it without feeling guilty of not studying? I think the pper should discuss this problem.

6. PLOS authors have the option to publish the peer review history of their article (what does this mean?). If published, this will include your full peer review and any attached files.

Reviewer #1: **Yes: **Jennifer Faux-Campbell Ph.D.

Reviewer #2: No

---

## [Author Response · Author response to Decision Letter 0]

14 Jun 2021

We thank the reviewers for their useful suggestions and recommendations! We provided the explanations and we described all the changes made to the document in a separate Word document entitled Response to Reviewers. We uploaded this document as a separate file.

---

## [Decision Letter · Decision Letter 1]

1 Jul 2021

PONE-D-21-10280R1

Dear student what should I write on my wall? A case study on academic uses of Facebook and Instagram during the pandemic

PLOS ONE

Dear Dr. Coman,

Thank you for submitting your manuscript to PLOS ONE. After careful consideration, we feel that it has merit but does not fully meet PLOS ONE’s publication criteria as it currently stands. Therefore, we invite you to submit a revised version of the manuscript that addresses the points raised during the review process.

The paper needs MINOR REVISIONS. The authors should revise the paper to address the suggestion highlighted by the reviewers.

We look forward to receiving your revised manuscript.

Kind regards,

Barbara Guidi

Academic Editor

PLOS ONE

Journal Requirements:

Reviewers' comments:

Reviewer's Responses to Questions

**Comments to the Author**

1. If the authors have adequately addressed your comments raised in a previous round of review and you feel that this manuscript is now acceptable for publication, you may indicate that here to bypass the “Comments to the Author” section, enter your conflict of interest statement in the “Confidential to Editor” section, and submit your "Accept" recommendation.

Reviewer #1: (No Response)

Reviewer #2: All comments have been addressed

2. Is the manuscript technically sound, and do the data support the conclusions?

Reviewer #1: Yes

Reviewer #2: Yes

3. Has the statistical analysis been performed appropriately and rigorously? 

Reviewer #1: Yes

Reviewer #2: Yes

4. Have the authors made all data underlying the findings in their manuscript fully available?

Reviewer #1: Yes

Reviewer #2: Yes

5. Is the manuscript presented in an intelligible fashion and written in standard English?

Reviewer #1: No

Reviewer #2: Yes

6. Review Comments to the Author

Reviewer #1: Line 48

Minor formatting concern-should be another space between subtitle and text.

Line 57 traditional face- to – face courses, and also being perceived as platforms that can improve the

Just a minor comment-it looks like a different symbol was used or the font varies in the face-to-face section.

Line 95 between people [11] 96 Social networks have been defined by Boyd and Ellison, as web based services that offer

This paragraph seems too short. It like it, but you may want to incorporate it into its following paragraph.

Line 133 showed that 42, 17% of the

I think this is typo. Maybe you mean to say showed that 17% of the …?

Line 136 online communication, social networks helping them accommodate

Run on sentence, perhaps say …online communication, with social networks helping them accommodate easier to university life, and by…

Line 163 collaboration between students, by allowing them to chat in real time that facilitates effective feedback and 164 comments on the posts of their peers

Line 164 Social networks help teachers

Line 170 platforms such as Facebook registered lower grade-point averages,

Line 177 internet use, in January 2020 there were 3.8 billion social media users and Facebook, having 178 2.44 billion

Line 194 high interest for studying the effects of Facebook and what students use it for [55]. 195 According

Connect these two paragraphs

Line 201 students also use Facebook for discussing educational content, [57], while another research

Word research is becoming redundant. Perhaps say study for one instead.

Line 222 educational tool [66]. 11 223 Instagram is

Connect these two paragraphs

Line 254 The present cross

Minor formatting concern-should be another space between subtitle and text.

Line 272 Based on the data gleaned

Minor formatting concern-should be another space between subtitle and text.

Line 293 The data was analyzed

Minor formatting concern-should be another space between subtitle and text.

Line 354 The means (Table 2) indicate that most

Minor formatting concern-should be another space between subtitle and text.

Line 418 In consideration to the use of social

Minor formatting concern-should be another space between subtitle and text.

Line 443 In consideration to the use of Facebook

Minor formatting concern-should be another space between subtitle and text.

Line 462 Previous studies on social networks use

Minor formatting concern-should be another space between subtitle and text.

Line 472 research that could complement and come as a response to a similar study

Could complement and act as a response to a similar study

Line 474 research that could come as a response to a previous study

Could support a previous study

Also, more information about this previous study. Who conducted it? For what purpose? What were the results?

Line 485 Even more, to highlight why we focused only on Facebook and Instagram, and not on other 486 platforms such as Yoututbe

Further, we focused solely on Facebook and Instagram…also should be YouTube

Line 487 YouTube

Line 489 with diversified content, we considered

With diversified content. As a consequence, we considered….

Line 491 between students. As far as Tik Tok is concernced, being a relatively new platform, we 492 considered that the analysis of the way it could be integrated into the educational process, 493 could have altered the replicability of the research, because the development and use of Tik 25 494 Tok globally is rather uneven.

In regard to Tik Tok, while we considered the educational benefits of the platform, we decided to exclude it from the study because it is a relatively new platform and people’s response to the application is uneven.

Line 500 Researchgate is a platform that aims to 500 promote research, to spread knowledge through scientific papers, and we wanted to focus on 501 social networks that offer more diverse functions.

While Researchgate is a beneficial platform for promoting research and spreading knowledge trough the scientific community, it is not a widely known application with diverse functions.

Line 502 that could be used by students and teachers for searching for jobs and internships, but our 503 interest was in finding out how social networks more general and comprehensive,

Internships; however, our interest

Line 505 Furthermore, Coursersa was not in 505 our area of interest because the focus of this paper was on social networks, on platforms that 506 were not initially designed for educational purposes

Finally, Coursersa was not a focus of this particular study because it centralized its format specifically on educational purposes, rather than social network outreach.

Line 510 they allow students to properly communicate and collaborate with eachother during courses

Each other

Line 511 for example, on Facebook, teachers could create groups and students could work in teams in 512 order to fulfil certain tasks, that can later be presented to the other teams on a live meeting)

Separate sentence entirely

Line 513 and last but not least, students are familiar with these social networks, and even though they 514 usually use them to relax and communicate with their friends, because they are already 26 515 accustomed to them, students may be more relaxed, more creative and more willing to learn in 516 a familiar environment.

Separate sentence entirely

Line 519 used to promote knowledge in higher education, but only 26,7%

26.7%

Line 536 educational process, students would rather use Facebook. Thus,

You use “thus” too frequently. You can use “as a consequence” instead.

Line 535 According to our findings, if they were to use Facebook and Instagram in the 536 educational process, students would rather use Facebook. Thus, this higher preference for 27 537 Facebook, and the idea that Facebook is the most suitable platforms for students, might also be 538 influenced by the fact that the platform was initially created for students [61], in order to help 539 them socialize. A

Very choppy. Consider re-wording these sentences.

For example, “according to our findings, students prefer using Facebook exclusively for educational purposes, opting to exclude the use of Instagram. Because students prefer Facebook, and because it is better suited for students academically, this platform may aid instructors in student socialization and interaction.

Another aspect revealed by our research is that master’s and Ph.D. students are 540 more willing to use Facebook for educational purposes than, Bachelor students. Thus, this 541 result is in line with other studies conducted in Romania, [8] which showed that Master 542 students were fonder of using online platforms, than Bachelor students.

“our study also revealed acceptance of these social platforms by master’s and Ph.D. students, which concurs with previous studies conducted in Romania which supported the use of online platforms in students with advanced educational degrees as opposed to Bachelor’s students.”

Line 548 process. 549 In the case of Instagram, i

Combine these paragraphs

Line 551 students who use daily their Facebook and Instagram accounts

Students who use their Facebook and Instagram accounts daily

Line 557 such as Facebook and Instagram is relevant and necessary. Thus

Again, thus

Line 569 respondents daily used Facebook for more than 4,5 hours

4.5 hours

Line 584 Taking into consideration the aspects mentioned above, certain issues can be 585 addressed. The first issue refers to whether Moodle platforms

Not all educators are versed in Moodle. You should define this platform

Line 594 these platforms only because they would have a reasonable motive for using them, thus not 595 developing a feeling of guilt for not studying. Thus, a

Thus, again

Line 606 In the context

Minor formatting concern-should be another space between subtitle and text.

Line 629 Taking into account the previously mentioned aspects, some theoretical and practical 630 implications of our paper can be highlighted. Considering the

Awkward sentence structure. Consider something like “turning to the theoretical and practical implications of our paper, this research can contribute to the literature…

Line 639 Therefore

Minor formatting concern-should be another space between subtitle and text.

Also, delete therefore

Line 640 future researches. Our study being exploratory

Because our study is exploratory, the instruments

Reviewer #2: I think the authors made an excellent work in addressing all the comments.

I only have one final suggestion: I understand Steemit is not as popular as other platforms, so maybe a citation that explains its features is due to help the reader in case she/he wants to to delve into the platform. Just add the following reference the first time you name Steemit in your paper: https://ieeexplore.ieee.org/document/9298888/

7. PLOS authors have the option to publish the peer review history of their article (what does this mean?). If published, this will include your full peer review and any attached files.

Reviewer #1: **Yes: **Jennifer Faux-Campbell

Reviewer #2: No

---

## [Author Response · Author response to Decision Letter 1]

19 Jul 2021

We thank the reviewers very much for their useful recommendations. We adressed each recommendation and we described the changes made to the text in the separate file that we uploaded, which is entitled: Response to reviewers.

---

## [Decision Letter · Decision Letter 2]

9 Sep 2021

Dear student what should I write on my wall? A case study on academic uses of Facebook and Instagram during the pandemic

PONE-D-21-10280R2

Dear Dr. Coman,

We’re pleased to inform you that your manuscript has been judged scientifically suitable for publication and will be formally accepted for publication once it meets all outstanding technical requirements.

Kind regards,

Barbara Guidi

Academic Editor

PLOS ONE

Additional Editor Comments (optional):

Reviewers' comments:

Reviewer's Responses to Questions

**Comments to the Author**

1. If the authors have adequately addressed your comments raised in a previous round of review and you feel that this manuscript is now acceptable for publication, you may indicate that here to bypass the “Comments to the Author” section, enter your conflict of interest statement in the “Confidential to Editor” section, and submit your "Accept" recommendation.

Reviewer #2: All comments have been addressed

2. Is the manuscript technically sound, and do the data support the conclusions?

Reviewer #2: Yes

3. Has the statistical analysis been performed appropriately and rigorously? 

Reviewer #2: Yes

4. Have the authors made all data underlying the findings in their manuscript fully available?

Reviewer #2: Yes

5. Is the manuscript presented in an intelligible fashion and written in standard English?

Reviewer #2: Yes

6. Review Comments to the Author

Reviewer #2: The authors did a great job imporivng the quality of the manuscript and I think the paper is finally ready for publication.

7. PLOS authors have the option to publish the peer review history of their article (what does this mean?). If published, this will include your full peer review and any attached files.

Reviewer #2: No

---

## [Editor Report · Acceptance letter]

14 Sep 2021

PONE-D-21-10280R2 

Dear student, what should I write on my wall?
A case study on academic uses of Facebook and Instagram during the pandemic 

Dear Dr. Coman:

I'm pleased to inform you that your manuscript has been deemed suitable for publication in PLOS ONE. Congratulations! Your manuscript is now with our production department. 

Kind regards, 

on behalf of

Dr. Barbara Guidi 

Academic Editor

PLOS ONE